# Skinfold Thickness Distribution in Recreational Marathon Runners

**DOI:** 10.3390/ijerph17092978

**Published:** 2020-04-25

**Authors:** Pantelis Theodoros Nikolaidis, Thomas Rosemann, Beat Knechtle

**Affiliations:** 1School of Health and Caring Sciences, University of West Attica, 12241 Egaleo, Greece; pademil@hotmail.com; 2Exercise Physiology Laboratory, 18450 Nikaia, Greece; 3Medbase St. Gallen Am Vadianplatz, 9001 St. Gallen, Switzerland; Thomas.Rosemann@usz.ch; 4Institute of Primary Care, University of Zurich, 8091 Zurich, Switzerland

**Keywords:** anthropometry, body composition, endurance exercise, long distance, skinfold caliper

## Abstract

The relationship of body fat (BF) percentage with performance of elite marathon runners has been well studied; however, less information is available about the variation of skinfold thickness by sex and performance in non-elite marathon runners. The aim of the present study was to examine the variation of skinfold thickness by sex and performance in recreational marathon runners. Participants included 32 female (age 40.1 ± 9.0 years, BF 19.6 ± 4.7%, and training volume 47.7 ± 22.6 km) and 134 male marathon runners (44.3 ± 8.8 years, 17.6 ± 4.0%, and 53.0 ± 21.2 km, respectively). The largest skinfold thickness was the abdomen in both sexes, whereas the smallest was biceps in men, and chins in women (*p* < 0.001). The largest sex difference in skinfold thickness was observed in triceps being the fattest in women (*p* < 0.001). The largest difference in skinfold thickness among men’s performance groups was observed in the iliac crest, and the smallest in the patella and proximal calf (*p* < 0.001). In summary, skinfold measurements indicated that women had more fat in both their upper and lower limbs, while men had more fat in their trunk. With regards to the role of performance level, the slowest runners presented relatively more fat in the upper limbs and trunk anatomical sites, i.e., away from the active muscles of legs.

## 1. Introduction

Considering the increased popularity of marathon running, a large number of studies have examined determinants of race time [1,2,3,4]. Nowadays, it is well known that marathon race time might be predicted by performance indices (e.g., 10 km race time, number of previously finished marathon races) [2], training volume and intensity [1,3], and physiological [5] and anthropometric characteristics (e.g., body mass index) [4]. In addition, marathon race time has been related to body composition [6], where a higher body fat (BF) percentage was associated with slower race time.

Studies on marathon runners have used several assessment methods of BF and other measures of body composition, such as muscle mass. BF has been widely assessed using skinfolds’ thickness in seven [6,7] and ten sites [8]. Not only the quantity, but also the distribution of fat has health implications, with central fat related to risk for diseases such as coronary heart disease and stroke, whereas peripheral fat has less metabolic risk [9]. In addition to health, the fat distribution might have implications for endurance performance, e.g., it was shown that slow mountain marathon race time was related with a high level of central adiposity [10]. With regards to the role of sex, male swimmers had a more central distribution of fat, whereas women had more fat in their lower limbs [11].

Although the abovementioned studies have enhanced our understanding of anthropometric characteristics and the body composition of marathon runners, little information exists so far about the variation of skinfold thickness by anatomical site. Since the number and location of anatomical sites used for skinfold thickness assessment in this sport varied by study [6,7,8], knowledge on the variation of skinfold thickness by anatomical site would be of great practical importance for practitioners to effectively monitor the body composition of their athletes. For instance, it is well-known that fast and slow marathon runners differed in BF estimated by the sum of skinfolds [1,12]; however, the anatomical sites that accounted for this variation have not been examined so far. Therefore, the aim of the present study was to examine the variation of skinfold thickness by sex and performance in recreational marathon runners.

## 2. Materials and Methods

### 2.1. Participants

Participants included 32 female (age 40.1 ± 9.0 years) and 134 male marathon runners (age 44.3 ± 8.8 years) who had finished the Athens marathon in 2017 (Table 1), i.e., the men-to-women ratio (MWR) was 4.19 in this study. The number of finishers in this race was 2915 female and 11,828 male runners, resulting in an MWR of 4.06 [13]. Despite the smaller number of female participants in the present study, the MWR in our sample might be considered similar to the overall observed in this race. Female and male participants had previously finished 3.3 ± 3.6 and 5.6 ± 6.3 marathon races, trained 4.1 ± 1.5 and 4.4 ± 1.2 days·wk^−1^, and covered 47.7 ± 22.6 and 53.0 ± 21.2 km·wk^−1^, respectively. After being informed about the procedures of the research, all participants provided written informed consent. The study design was in accordance to the Declaration of Helsinki revised in 2013 and was approved by the local Institutional Review Board (EPL 2017/3). Based on their personal best record, four performance groups were considered in men (<3:30 h:min, *n* = 32; 3:30–4:00 h:min, *n* = 33; 4:00–4:30 h:min, *n* = 36; >4:30 h:min, *n* = 33), and two groups in women (<4:30 h:min, *n* = 15; ≥4:30 h:min, *n* = 17). The number of performance groups for each sex was selected depending on their sample sizes.

### 2.2. Procedures

The present study was part of a larger project on physiological characteristics of recreational marathon runners, and detailed procedures were published elsewhere [11,13,14,15]. The data were collected about four weeks before the Athens marathon in 2017. Briefly, participants underwent a series of anthropometric and exercise tests. Height and weight were measured with subjects in minimal clothing and barefoot. A Tanita HD351 digital weighing scale (Tanita, Arlington Heights, IL, USA) was used for measurement of weight (to the nearest 0.1 kg), and a SECA 213 portable stadiometer (SECA, Leicester, UK) for height (0.1 cm). Body mass index was calculated as the quotient of weight (kg) to height squared (m^2^). Eleven skinfolds (cheek, chin, pectoral, biceps triceps, subscapular, abdomen, chest II, iliac crest, patella, and proximal calf) were examined for thickness on the right side of the body to the nearest 0.1 mm using a Harpenden skinfold caliper (Harpenden, West Sussex, UK) according to procedures described by Eston and Reilly [16].

### 2.3. Statistical and Data Analysis

Statistical analyses were conducted using GraphPad Prism v. 7.0 (GraphPad Software, San Diego, CA, USA) and IBM SPSS v.23.0 (SPSS, Chicago, IL, USA). Statistical significance was set at *p* ≤ 0.05. Data were tested for normality using the Kolmogorov–Smirnov test and visual inspection of Q-Q plots, and were normally distributed, suggesting the use of parametric statistics. Data were presented as mean and standard deviations. A between-within analysis of variance (ANOVA) examined the main effects of anatomical site and sex, and their interaction on skinfold thickness and eta squared (η^2^) estimated the magnitude of these differences. An independent Student t-test compared anthropometric characteristics and body composition between female and male participants, and between female performance groups. Cohen’s d examined the magnitude of differences in t-test.

## 3. Results

The anthropometric characteristics of participants were presented in Table 1. Overall, a large main effect of anatomical site on skinfold thickness was observed (*p* < 0.001, η^2^ = 0.516), with the smallest score in biceps (5.4 ± 2.8 mm) and the largest in abdomen (21.4 ± 8.3 mm). Particularly, in women, the smallest score was observed in chin (6.8 ± 2.7 mm), and the largest in abdomen (18.2 ± 6.4 mm; *p* < 0.001, η^2^ = 0.641). Whereas, in men, the smallest score was shown in biceps (5.1 ± 1.9 mm), and the largest in abdomen (22.2 ± 8.5 mm; *p* < 0.001, η^2^ = 0.700). A moderate sex × anatomical site interaction on skinfold thickness was shown (*p* < 0.001, η^2^ = 0.516), with the sex difference ranging from −34.0% (leaner pectoral in women) to 35.8% (fatter triceps in women) in Table 2.

No difference was shown between female performance groups (Table 3). In male participants, the largest magnitude of differences among performance groups was observed in iliac crest, followed by abdomen and pectoral skinfolds; whereas the smallest was shown in leg skinfolds (patella and proximal calf) (Table 4).

## 4. Discussion

The main findings of the present study were that (a) skinfold measures indicated that women had more fat in both upper and lower limbs skinfolds, while men had more fat in trunk skinfolds, (b) the largest sex difference in skinfold thickness was observed in triceps being larger in women, and (c) the largest difference in skinfold thickness among men’s performance groups was observed in the iliac crest, and the smallest in patella and proximal calf.

The higher BF observed in women compared to in men was in agreement with the existing literature in endurance runners [17]. For instance, it was observed in half-marathon runners that triceps, front thigh, and medial calf skinfold thicknesses were smaller in males runners, compared to female runners [18]. However, the quantification of sex differences in fat distribution was a novel finding showing different patterns; women had more fat in both upper and lower limbs skinfolds (triceps, biceps, patella, and proximal calf), whereas men had more fat in trunk skinfolds (pectoral, abdomen, and iliac crest).

The comparison of anthropometric characteristics among performance groups in men revealed that the faster runners had smaller skinfold thicknesses than their slower counterparts. This was in agreement with the notion that a faster race time was associated with smaller skinfold thickness. It was observed that marathon race speed correlated moderately with medial calf, mid-axilla, pectoral, front thigh, and suprailiac skinfold thickness in male runners [12]. In addition, the body composition profile might be unique for marathon runners, e.g., compared to ultra-marathon runners, male marathon runners had larger pectoral, mid-axillary, and suprailiac skinfold thicknesses [17]. In addition, our findings indicated larger differences among performance groups in skinfold sites distal, rather than proximal, to muscles of lower limbs performing the main locomotion. This observation indicated that running would be related to low skinfold thickness of lower limbs, attenuating the differences among performance groups. On the other hand, no difference in skinfolds was observed between the two performance groups in women, which may be attributed to their relatively small sample compared to men [19].

The superior body composition profile of the faster participants might be due to their sport experience (expressed by the number of finished marathon races) and training volume (weekly training sessions, and distance covered during a week). This large training volume accounted for a large part of the variance of BF, as it was shown that increased fatty acid oxidation occurred during submaximal and prolonged exercise [20]. Although nutrition was not considered in the present study, it might be assumed that fast participants would adopt better nutrition and supplementation strategies [21]. Moreover, the lowest level of skinfold thickness in the faster participants highlighted the role of BF on sport performance. This finding was also in agreement with studies in other endurance sports (e.g., rowing [22] and cycling [23]), speed disciplines [24], combat sports, [25] and power-related exercises [26], where lower BF was associated with superior performance.

A limitation of our study was the use of a specific assessment method for skinfold thickness. It was acknowledged that there were other skinfold thickness methods, mostly using a smaller number of skinfolds, that would provide different estimates of BF [27]. Thus, caution would be needed to generalize the findings of this study to other assessment methods. The strength of the study was the use of a skinfold assessment method with many anatomical sites [16], allowing the detailed study of fat distribution. Considering the large number of recreational runners competing in marathon races [28,29] and the role of BF on race time [1,12], the findings would have practical applications in the context of monitoring training. Among the skinfold sites, practitioners should be advised to monitor anatomical sites presenting the largest variation, such as the pectoral, abdomen, and iliac crest. Future studies should examine the fat distribution assessed by the skinfold method with regards to more valid measures of BF, such as the ultrasound technique [30].

## 5. Conclusions

In summary, women were fatter in both upper and lower limbs skinfolds (triceps, biceps, patella, and proximal calf), whereas men were fatter in trunk skinfolds (pectoral, abdomen, and iliac crest). With regards to the role of performance level, the slowest runners presented relatively more fat in upper limbs and trunk anatomical sites, i.e., away from the active muscles of legs.

## Figures and Tables

**Table 1 ijerph-17-02978-t001:** Anthropometric characteristics and body composition by sex.

Variable	Women (*n* = 32)	Men (*n* = 134)	Cohen’s d
Age (years)	40.1 ± 9.0	44.3 ± 8.8 *	−0.47
**Anthropometry**			
Height (cm)	162.3 ± 6.5	176.1 ± 5.8 **	−2.24
Body mass (kg)	57.7 ± 7.5	76.8 ± 9.2 **	−2.28
BMI (kg·m^−2^)	21.8 ± 2.2	24.7 ± 2.6 **	−1.20
BF (%)	19.6 ± 4.7	17.7 ± 4.0 *	0.44

BMI = body mass index, BF = body fat percentage, * *p* < 0.05, ** *p* < 0.001.

**Table 2 ijerph-17-02978-t002:** Skinfold thickness by sex.

Skinfold	Women (*n* = 32)	Men (*n* = 134)	%Difference	*p*	Cohen’s d
Cheek (mm)	7.6 ± 1.7	8.0 ± 1.9	−5.5	0.250	−0.22
Chin (mm)	6.8 ± 2.7	6.8 ± 2.1	0.7	0.915	0
Triceps (mm)	13.5 ± 4.0	8.7 ± 2.9	35.8	<0.001	1.37
Subscapular (mm)	13.5 ± 5.3	13.6 ± 5.0	−0.8	0.918	−0.02
Pectoral (mm)	7.7 ± 3.4	10.3 ± 5.6	−34.0	0.012	−0.56
Chest II (mm)	11.5 ± 3.9	11.4 ± 4.6	0.8	0.921	0.02
Abdomen (mm)	18.2 ± 6.4	22.1 ± 8.4	−21.3	0.016	−0.52
Iliac crest (mm)	14.9 ± 5.8	18.0 ± 7.1	−20.9	0.022	−0.48
Patella (mm)	13.0 ± 3.4	9.9 ± 2.9	23.4	<0.001	0.98
Proximal calf (mm)	10.6 ± 3.4	7.2 ± 2.5	31.7	<0.001	1.14
Biceps (mm)	7.0 ± 3.0	5.0 ± 1.9	27.9	<0.001	0.80

**Table 3 ijerph-17-02978-t003:** Anthropometric characteristics and body composition of female participants by performance level.

Variable	Performance Group		*p*	Cohen’s d
	<4:30 h:min (*n* = 15)	≥4:30 h:min (*n* = 17)		
Finished marathons (*n*)	4.2 ± 4.8	2.5 ± 2.1	0.199	0.46
Training days (wk^−1^)	4.4 ± 1.7	3.8 ± 1.2	0.221	0.41
Training distance (km·wk^−1^)	55.2 ± 23.2	40.7 ± 20.4	0.096	0.66
BF (%)	20.7 ± 2.9	18.6 ± 5.7	0.204	0.46
*Skinfolds*				
Cheek (mm)	8.1 ± 1.7	7.2 ± 1.6	0.132	0.55
Chin (mm)	7.4 ± 2.7	6.3 ± 2.6	0.250	0.42
Triceps (mm)	13.9 ± 2.5	13.2 ± 5.0	0.660	0.18
Subscapular (mm)	13.1 ± 3.8	13.8 ± 6.5	0.738	−0.13
Pectoral (mm)	7.5 ± 2.2	7.8 ± 4.3	0.828	−0.09
Chest II (mm)	11.1 ± 2.5	11.8 ± 5.0	0.665	−0.18
Abdomen (mm)	19.0 ± 4.5	17.5 ± 7.8	0.875	0.24
Iliac crest (mm)	15.1 ± 3.8	14.8 ± 7.2	0.875	0.05
Patella (mm)	13.8 ± 3.0	12.2 ± 3.6	0.197	0.48
Proximal calf (mm)	11.0 ± 3.0	10.2 ± 3.7	0.530	0.24
Biceps (mm)	7.7 ± 3.3	6.4 ± 2.6	0.231	0.44

BF = body fat percentage.

**Table 4 ijerph-17-02978-t004:** Anthropometric characteristics and body composition of male participants by performance level.

Variable	Performance Group				*p*	η^2^
	<3:30 h:min (*n* = 32)	3:30–4:00 h:min (*n* = 33)	4:00–4:30 h:min (*n* = 36)	>4:30 h:min (*n* = 33)		
Finished marathons (*n*)	7.7 ± 6.2	8.9 ± 9.5	3.6 ± 2.5	2.5 ± 1.6	<0.001	0.182
Training days (wk^−1^)	5.3 ± 1.2	4.5 ± 1.2	4.1 ± 0.7	3.6 ± 1.1	<0.001	0.261
Training distance (km·wk^−1^)	68.0 ± 23.7	58.2 ± 20.7	45.7 ± 12.3	40.4 ± 15.8	<0.001	0.254
BF (%)	14.2 ± 3.9	18.1 ± 3.6	18.0 ± 2.9	20.0 ± 3.5	<0.001	0.273
Skinfolds						
Cheek (mm)	7.1 ± 1.3	8.3 ± 2.1	7.7 ± 1.6	8.9 ± 2.0	0.001	0.128
Chin (mm)	5.6 ± 1.5	7.1 ± 2.4	6.6 ± 1.6	7.6 ± 2.3	0.001	0.121
Triceps (mm)	7.4 ± 2.7	8.6 ± 2.4	8.6 ± 2.8	10.0 ± 2.7	0.003	0.113
Subscapular (mm)	10.9 ± 3.9	13.4 ± 5.0	13.5 ± 3.8	16.5 ± 6.1	<0.001	0.151
Pectoral (mm)	6.4 ± 2.9	10.6 ± 5.4	9.9 ± 4.3	14.1 ± 6.6	<0.001	0.236
Chest II (mm)	8.5 ± 3.2	11.5 ± 4.4	11.6 ± 4.0	13.8 ± 5.2	<0.001	0.160
Abdomen (mm)	15.3 ± 6.9	22.9 ± 7.7	22.6 ± 6.3	27.1 ± 8.3	<0.001	0.241
Iliac crest (mm)	12.5 ± 6.1	18.0 ± 6.1	18.5 ± 5.6	22.7 ± 6.8	<0.001	0.256
Patella (mm)	9.0 ± 2.5	9.6 ± 2.5	9.7 ± 3.0	11.1 ± 3.0	0.026	0.072
Proximal calf (mm)	6.2 ± 2.4	7.3 ± 2.0	7.2 ± 2.4	7.9 ± 2.8	0.050	0.061
Biceps (mm)	4.0 ± 1.3	5.0 ± 2.1	5.1 ± 1.5	5.9 ± 2.1	<0.001	0.134

BF = body fat percentage.

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
