# Peer review of "Skinfold Thickness Distribution in Recreational Marathon Runners"

_ijerph, 2020, doi:10.3390/ijerph17092978_

Round 1

Reviewer 1 Report

Please find attached "comments and suggestions for the authors".

Author Response

Reviewer 1

31-32 When stating number of studies, authors should provide at least 3 studies (not only

1 reference) number means more than 1.

Answer: We agree with the expert reviewer and corrected it as suggested.

38-39 Change to in seven (6,7) and ten (8) sites.

Answer: We agree with the expert reviewer and corrected it as suggested.

61 Since the participants were tested, what was their height and weight? what was

their records number in marathon. what level are they competing on etc... a

complete information regarding participants should be provided. alternatively,

authors could point to table 1 for the full description of anthropometry and table 3

for performance level. however, this being said, Table 3 is confusing as it is unclear

what it represents e.g., all participants, males?? and not female and male separately.

this needs to be rethought and considered.

Answer: We agree with the expert reviewer and

78 Advised to use p 0.05

Answer: We agree with the expert reviewer and corrected it as suggested.

Table 1 Change diameter to circumference .

Answer: We agree with the expert reviewer and corrected it.

113 in bland Altman analysis, the t-test difference of 3.9±2.7% is what so called Bias or

(systematic error). since the authors are looking at agreement between the methods,

what was the accepted level of agreement between the methods? there no two

instrument would give 100% identical results, therefore, for the method to be

useful, the authors should provide the accepted level of agreement (meaning, the

allowed error).

Answer: We agree with the expert reviewer and added this information in the methods and results (“To evaluate the acceptable level of agreement between skinfold and BIA method’s BF, the minimal detectable change in BF was considered, which in the case of BIA was 1kg of fat mass [17] corresponding to BF 1.7% in female and 1.3% in male participants.”).

So, to say, the authors should inform the reader about the systematic and the

random variation and the total error accepted. I would advise the authors to

thoroughly read the following paper as it represents the case in a very simple

manner:

Validity and reliability of the Newtest Powertimer 300-series® testing system.

https://doi.org/10.1080/02640410802448723

Answer: We agree with the expert reviewer and added this aspect (see previous answer).

It should be further noted that the results presented is 95% CI. It should be 95%

limits of agreement instead.

Answer: We agree with the expert reviewer and corrected it as suggested.

115-117 Does this mean that the error was systematic? furthermore, in the statistical analysis

the authors named the testing of normality, was the data normally distributed?

Answer: We agree with the expert reviewer and added this information in the Results (“…magnitude (r=-0.23, p=0.009) in men, i.e. the higher the BF, the smaller their difference, whereas no correlation was found in women (r=-0.13, p=0.465)”) and Methods (“and were normally distributed suggesting the use of parametric statistics”).

After reading the discussion could immediately note that the (a) the variation of body

composition of marathon runners by sex and performance level, and (b) the agreement between

measures relied on skinfols’ thichness and BIA to evaluate fat and muscle mass. Were not

fully addressed. First, the performance discussion is limited to only male participants (see table

3). Point B was not discussed and only named in the limitation of the study. The authors further

should distinguish between gender and not discuss generally as their results did not entirely

present for both genders.

In general, a great work is need in the discussion, so it covers the purpose of the study.

Answer: We agree with the expert reviewer and added relevant table for women and revised a large part of the discussion adding a paragraph on the comparison of the two BF assessment methods and discussing the findings specifically for each gender.

Reviewer 2 Report

Review of the article

Body composition and fat distribution in recreational marathon runners” (ijerph-752924)

The manuscript entitledBody composition and fat distribution in recreational marathon runners” (ijerph-752924) is well written and raises very important aspects related to the importance of body composition in sport, especially for marathon runners.

However, it contains several important points, which, according to the reviewer's opinion, are a significant weakness of this work. Increasing the quality of this work requires significant reorganization of this manuscript.

Individual specific comments are given below.

GENERAL REMARKS

  1. The work concerns the important aspects as mentioned (body composition - sport - performance), but finally the Authors did not present anything new or revealing. These studies are not original and do not add much value in their current form. The fact that body composition differs between men and women in distribution and quantity is obvious and requires no research.
  2. As mentioned above for example, the conclusions (lines 173-177) of the work are trivial and obvious. It didn't need to do any studies to find out that (a)"different patterns of fat distribution were observed in women and men” or (b) “the slowest runners presented relatively more fat..”.
  3. In the work, part of the results divided depending on the duration of the marathon (this is certainly a good idea), but the Authors did not show the essence of the problem. Once again, it is obvious and well known that athletes/competitors with higher training experience / athletic level have simultaneously "better" body composition (lower BF, and higher % of FFM). This differences are also related to the greater volume of training, number of training sessions per week/day, experience and training experience, often a different diet and supplementation, recovery and physiotherapy methods, etc .. This work lacks the above-mentioned key information that is necessary to properly follow and interpret results.
  4. It has not been fully demonstrated, nor has it been comprehensively discussed, that body composition is significantly related to physical capacity / performance in various sports.
    This has been reported in several papers, including in endurance disciplines (such as rowing: J Sports Med Phys Fitness. 2019, 59, 1526-1535, cycling: Nutr Hosp. 2015, 32, 2223-7) or speed-strength disciplines (e.g. sprint running (J Sports Med Phys Fitness. 2017, 57, 1142-1146), combat sports (Archives of Budo, 2016, 12, 247-256) and power-related exercises (Int J Sports Physiol Perform. 2018, 13, 189-193.).
  5. A serious problem at work is the fact that very important aspect like performance, was shown very marginally and only men's data are included. This disturbs the sense of including women's results here due to female runners could have belonged to a group with very low/low performance. In addition, the number of these groups varied enormously (32F vs. 134M). For the aforementioned reasons, it is therefore not known whether comparing the body composition of these groups (Male / Female) has any sense whatsoever
  6. The differences between the methods used could be due to different levels of hydration. This would explain the higher differences in men who were also better hydrated. This is quite important. The weakness of this study is also the fact that no standard methods (e.g. DEXA / Bodpod) was done, because it is not known in practice which results of which method were the most truthful..

====

ABSTRACT

  1. There is no data on how many men / women participated, etc .. The range of age, body fat and training experience would also be useful, because it is not known whether the participants were homogeneous or very different group

METHODS

  1. This section lacks key information characterizing the study group. This is necessary for further interpretation of the results. Data like t training experience, number of trainings / distances run per week, if and how often they started should be added. Furthermore, was this research in the start or preparation period, or any specific training cycle etc.?
  2. Statement in section 2.2. (Procedures) that "The present study was part of a larger project on physiological characteristics of recreational marathon runners and detailed procedures were published elsewhere [12-15]" is hardly understandable. Analyzing the range/scope of this works and the results presented in manuscript, it is difficult to say whether it is "salami slices" or a new group? What are the novelty here? Furthermore, looking at the cited works (12-15), these groups and the specifics seem very different. Was all this analyzed within one "larger project"?
  3. The authors did not finally indicate how BF was measured by the BIA method. Probably Tanita?Because only "weight" is marked.Have all the recommendations been met for carrying out tests with this method / analyzer?What recommendations were based on?

RESULTS

  1. I have already described this problem partly in point 5. Why was no data given for women?Since comparisons between the sexes above are made, the body composition distribution of women related to performance should also be given.

DISCUSSION

  1. Again – authors underlined in the first sentence of the Discussion section that “The main findings of the present study were that (a) the largest skinfold thickness was abdomen in both sexes, whereas the smallest was biceps in men...”. This "main findings" is not revealing.
  2. Line 138 – authors wrote: “The higher BF observed in women than in men was in agreement with the existed literature in endurance runners..” - Again, this is evident not only in marathons, but in both sport and the typical population.

CONCLUSIONS

  1. Line 174 – the first sentence of the conclusions: “In summary, different patterns of fat distribution were observed in women and men.” - such a statement is trivial and presents an obviousness that did not require scientific research ...
  2. Lines 175-176 – the second sentence of the conclusions: “With regards to the role of performance level, the slowest runners presented relatively more fat in anatomical sites away from the active muscles of legs” - this is unfortunately also obvious. What's more, both of these aspects can result from differences in training etc ..
  3. Lines 176-177 – authors wrote: "Furthermore, practitioners working with marathon runners should be aware that BIA might provide lower BF scores than skinfold thickness method in men." - Generally, for the coach / practitioner, the credibility of indications induced by training changes will be more important than a single measurement.

REFERENCES

  1. Carelessly done section .. Necessary thorough editing and unification. Several records lack the Journals name and/or volume number (e.g. 7,9,10,11,12); Letter case have to be corrected in some records (e.g. Journals titles)

Author Response

Reviewer 2

Review of the article

“Body composition and fat distribution in recreational marathon runners” (ijerph-752924)

The manuscript entitled “Body composition and fat distribution in recreational marathon runners” (ijerph-752924) is well written and raises very important aspects related to the importance of body composition in sport, especially for marathon runners.

However, it contains several important points, which, according to the reviewer's opinion, are a significant weakness of this work. Increasing the quality of this work requires significant reorganization of this manuscript.

 Answer: We agree with the concerns of the expert reviewer and addressed them accordingly. Please, find our answers to the specific comments below and the changes within text highlighted in red.

Individual specific comments are given below.

GENERAL REMARKS

The work concerns the important aspects as mentioned (body composition - sport - performance), but finally the Authors did not present anything new or revealing. These studies are not original and do not add much value in their current form. The fact that body composition differs between men and women in distribution and quantity is obvious and requires no research.

Answer: We agree with the expert reviewer and developed the novel aspects of the study (see next answer).

As mentioned above for example, the conclusions (lines 173-177) of the work are trivial and obvious. It didn't need to do any studies to find out that (a)"different patterns of fat distribution were observed in women and men” or (b) “the slowest runners presented relatively more fat..”.

Answer: We agree with the expert reviewer and revised conclusions (“…women were fatter in both upper and lower limbs skinfolds (triceps, biceps, patella and proximal calf), whereas men were fatter in trunk skinfolds (pectoral, abdomen and iliac crest). With regards to the role of performance level, the slowest runners presented relatively more fat in upper limbs and trunk anatomical sites, i.e.,…”).

In the work, part of the results divided depending on the duration of the marathon (this is certainly a good idea), but the Authors did not show the essence of the problem. Once again, it is obvious and well known that athletes/competitors with higher training experience / athletic level have simultaneously "better" body composition (lower BF, and higher % of FFM). This differences are also related to the greater volume of training, number of training sessions per week/day, experience and training experience, often a different diet and supplementation, recovery and physiotherapy methods, etc .. This work lacks the above-mentioned key information that is necessary to properly follow and interpret results.

Answer: We agree with the expert reviewer and added this information in the discussion (“The superior body composition profile of the fast participants might be due to their sport experience (expressed by the number of finished marathon races) and training volume (weekly training sessions, distance covered during a week). This large training volume accounted for by a large part of the variance of BF, as it has been shown that increased fatty acid oxidation occurred during submaximal and prolonged exercise [PMID: 29344008]. Although nutrition was not considered in the present study, it might be assumed that fast participants would adopt a better nutrition and supplementation.”).

It has not been fully demonstrated, nor has it been comprehensively discussed, that body composition is significantly related to physical capacity / performance in various sports.
This has been reported in several papers, including in endurance disciplines (such as rowing: J Sports Med Phys Fitness. 2019, 59, 1526-1535, cycling: Nutr Hosp. 2015, 32, 2223-7) or speed-strength disciplines (e.g. sprint running (J Sports Med Phys Fitness. 2017, 57, 1142-1146), combat sports (Archives of Budo, 2016, 12, 247-256) and power-related exercises (Int J Sports Physiol Perform. 2018, 13, 189-193.).

Answer: We agree with the expert reviewer and added this aspect including the recommended references (“Moreover, the lowest level of skinfold thickness in the faster participants highlighted the role of BF on sport performance. This finding was also in agreement with studies in other endurance sports (e.g., rowing [PMID: 31610640] and cycling [PMID: 26545681]) as well as speed disciplines [PMID: 28085130], combat sports [Durkalec2016] and power-related exercise [PMID: 28530517], where lower BF was associated with superior performance.”).

A serious problem at work is the fact that very important aspect like performance, was shown very marginally and only men's data are included. This disturbs the sense of including women's results here due to female runners could have belonged to a group with very low/low performance. In addition, the number of these groups varied enormously (32F vs. 134M). For the aforementioned reasons, it is therefore not known whether comparing the body composition of these groups (Male / Female) has any sense whatsoever

Answer: We agree with the expert reviewer and developed these aspects in the methods where it is showed that the men-to-women ratio (MWR) in our sample is ecologically valid (“i.e. the men-to-women ratio was 4.19, who had finished the Athens marathon in 2017. The number of finishers in this race was 2,915 female and 11,828 male runners resulting in a MWR 4.06 [https://www.athensauthenticmarathon.gr/site/index.php/en/results-en/491-results-2017-marathon]. Despite the smaller number of female participants in the present study, the MWR in our sample might be considered similar as the overall observed in this race.”).

The differences between the methods used could be due to different levels of hydration. This would explain the higher differences in men who were also better hydrated. This is quite important. The weakness of this study is also the fact that no standard methods (e.g. DEXA / Bodpod) was done, because it is not known in practice which results of which method were the most truthful..

Answer: We agree with the expert reviewer and developed these aspects in the discussion section (“The comparison between the BIA and skinfold method showed similar results in women, but different in men, where BIA had the lowest score. The difference between these methods in men might be due to different level of hydration [29]. Moreover, the reliability of BIA varied by factors related to the instrument itself, including electrodes, operator, subject and environment [30]. The lower value by ~4% of men’s BF in BIA than in skinfold method was in agreement with previous studies showing respective differences in soldiers (~3.5%) [31] and physically active adults (~5.5%) [32].”) and limitations (“Future studies should examine the fat distribution assessed by skinfold method with regards to more valid measures of BF such as the ultrasound technique [33].”).

====

ABSTRACT

There is no data on how many men / women participated, etc .. The range of age, body fat and training experience would also be useful, because it is not known whether the participants were homogeneous or very different group.

Answer: We agree with the expert reviewer and added this information “Participants were 32 female (age 40.1±9.0 years, skinfold-estimated BF 19.6±4.7% and training volume 47.7±22.6km) and 134 male marathon runners (44.3±8.8 years, 17.6±4.0% and 53.0±21.2km, respectively).”.

METHODS

This section lacks key information characterizing the study group. This is necessary for further interpretation of the results. Data like t training experience, number of trainings / distances run per week, if and how often they started should be added. Furthermore, was this research in the start or preparation period, or any specific training cycle etc.?

Answer: We agree with the expert reviewer and added this information in the methods (“Female and male participants had finished previously 3.3±3.6 and 5.6±6.3 marathon races, trained 4.1±1.5 and 4.4±1.2 days.wk-1 and covered 47.7±22.6 and 53.0±21.2km.wk-1, respectively.”…“The data were collected about four weeks before the Athens marathon in 2017.”).

Statement in section 2.2. (Procedures) that "The present study was part of a larger project on physiological characteristics of recreational marathon runners and detailed procedures were published elsewhere [12-15]" is hardly understandable. Analyzing the range/scope of this works and the results presented in manuscript, it is difficult to say whether it is "salami slices" or a new group? What are the novelty here? Furthermore, looking at the cited works (12-15), these groups and the specifics seem very different. Was all this analyzed within one "larger project"?

Answer: We agree with the expert reviewer about this concert and revised this part. The novelty here is that this topic has not been examined in the previous papers. Particularly, this paper is based on skinfolds that have not been reported in any of our previous papers. Considering the length of the present paper, it can be seen that the content cannot “fit” together with more topics. This is the only one study of this project analyzing the values of each skinfold separately and examining the distribution of fat.

The authors did not finally indicate how BF was measured by the BIA method. Probably Tanita?Because only "weight" is marked.Have all the recommendations been met for carrying out tests with this method / analyzer?What recommendations were based on?

Answer: We agree with the expert reviewer and added this information in the methods (“In addition, BF was also estimated by BIA method using Tanita 545 (Tanita, Arlington Heights, IL, USA). To ensure accuracy using this BIA equipment, the guidelines of manufacturer were followed, where participants were instructed to abstain from abstain from eating for three hours prior to testing.”).

RESULTS

I have already described this problem partly in point 5. Why was no data given for women? Since comparisons between the sexes above are made, the body composition distribution of women related to performance should also be given.

Answer: We agree with the expert reviewer and added the suggested analysis in women.

DISCUSSION

Again – authors underlined in the first sentence of the Discussion section that “The main findings of the present study were that (a) the largest skinfold thickness was abdomen in both sexes, whereas the smallest was biceps in men...”. This "main findings" is not revealing.

Answer: We agree with the expert reviewer and changed the (a) to “women were fatter in both upper and lower limbs skinfolds, whereas men were fatter in trunk skinfolds”.

Line 138 – authors wrote: “The higher BF observed in women than in men was in agreement with the existed literature in endurance runners..” - Again, this is evident not only in marathons, but in both sport and the typical population.

Answer: We agree with the expert reviewer and developed this part to show the novelty (i.e. that some skinfolds are largest in women, but other are largest in men) (“However, the quantification of sex differences in fat distribution was a novel finding showing different patterns; women were fatter in both upper and lower limbs skinfolds (triceps, biceps, patella and proximal calf), whereas men were fatter in trunk skinfolds (pectoral, abdomen and iliac crest).”).

CONCLUSIONS

Line 174 – the first sentence of the conclusions: “In summary, different patterns of fat distribution were observed in women and men.” - such a statement is trivial and presents an obviousness that did not require scientific research ...

Answer: We agree with the expert reviewer and revised this part.

Lines 175-176 – the second sentence of the conclusions: “With regards to the role of performance level, the slowest runners presented relatively more fat in anatomical sites away from the active muscles of legs” - this is unfortunately also obvious. What's more, both of these aspects can result from differences in training etc ..

Answer: We agree with the expert reviewer and revised this part.

Lines 176-177 – authors wrote: "Furthermore, practitioners working with marathon runners should be aware that BIA might provide lower BF scores than skinfold thickness method in men." - Generally, for the coach / practitioner, the credibility of indications induced by training changes will be more important than a single measurement.

Answer: We agree with the expert reviewer and revised this part.

REFERENCES

Carelessly done section .. Necessary thorough editing and unification. Several records lack the Journals name and/or volume number (e.g. 7,9,10,11,12); Letter case have to be corrected in some records (e.g. Journals titles)

 Answer: We agree with the expert reviewer and revised this section. We apologize for the presentation of references in the previous form, which was due to the automatic references program that now is removed.

Round 2

Reviewer 1 Report

Comments and Suggestions for Authors are attached as pdf.

Author Response

The answers to the comments of the reviewer are presented within notes of the pdf file.

Reviewer 2 Report

The revised manuscript was altered in accordance with provided suggestions.

My two comments concern the following aspects:

  1. Line 66 - Replace "commas" with "dots" in numeric data “2,915 female and 11,828”
  2. Line 75 – inconsistencies can be seen in the inserted time periods for women – in “Participants” section authors wrote: “3:30 h:min, n=15; ≥3:30 h:min).” but in the “Results” section (Table 3) other times are included “<4:30 h:min (n=15) ≥4:30 h:min”. It should be corrected in line with the facts.

Author Response

Reviewer 2

The revised manuscript was altered in accordance with provided suggestions.

My two comments concern the following aspects:

  1. Line 66 - Replace "commas" with "dots" in numeric data “2,915 female and 11,828”

Answer: These numbers express thousands; thus, using dots would convert them to decimals. Alternatively, we do not use any symbol so it reads “2915 female and 11828”. Please, find the changes highlighted in green.

  1. Line 75 – inconsistencies can be seen in the inserted time periods for women – in “Participants” section authors wrote: “3:30 h:min, n=15; ≥3:30 h:min).” but in the “Results” section (Table 3) other times are included “<4:30 h:min (n=15) ≥4:30 h:min”. It should be corrected in line with the facts.

Answer: We agree with the expert reviewer and corrected the numbers in the “Participants” section to “(<4:30 h:min, n=15; ≥4:30 h:min, n=17)”.

Round 3

Reviewer 1 Report

General comments:

I would suggest that the authors consider taking out the method comparison from the study as it is weakening this study. while the study is very interesting and have a merit, this method comparison takes out all what is good in this study.

First, the argument for method comparison in this study was not strong scientifically, considering the fact that the authors compare homemade Tanita to Hardened which is gold standard. Second the main purpose of this article is to report the Physiological profile of those participants (which is very interesting).

Reliability of the systems was not reported; therefore, method comparison is not important and useless. When the one does a method comparison, the one need in fact to look if a system (Tanita) has an agreement with a gold standard (Hardened). The fact that this sort of criterion validity is not valid without measuring the reliability of the systems.

I see this study as a wonderful study if the authors take out all DIA data from it (very little) and methods comparison. Detailed challenges are attached.

Author Response

Answers to reviewer’s comments

General comments:

I would suggest that the authors consider taking out the method comparison from the study as it is weakening this study. while the study is very interesting and have a merit, this method comparison takes out all what is good in this study.

First, the argument for method comparison in this study was not strong scientifically, considering the fact that the authors compare homemade Tanita to Hardened which is gold standard. Second the main purpose of this article is to report the Physiological profile of those participants (which is very interesting).

Reliability of the systems was not reported; therefore, method comparison is not important and useless. When the one does a method comparison, the one need in fact to look if a system (Tanita) has an agreement with a gold standard (Hardened). The fact that this sort of criterion validity is not valid without measuring the reliability of the systems.

I see this study as a wonderful study if the authors take out all DIA data from it (very little) and methods comparison. Detailed challenges are attached.

Answer: We agree with the expert reviewer and deleted the aspect of BIA throughout the paper and the role of age. Please, find the answers in the specific comments within the attached pdf file and the changes within the text highlighted in blue. Accordingly, we changed the title, deleted the figures and reduced the number of references.